# DL-Based Physical Tamper Attack Detection in OFDM Systems with Multiple Receiver Antennas: A Performance–Complexity Trade-Off

**DOI:** 10.3390/s22176547

**Published:** 2022-08-30

**Authors:** Eshagh Dehmollaian, Bernhard Etzlinger, Núria Ballber Torres, Andreas Springer

**Affiliations:** 1JKU LIT SAL eSPML Lab, Institute for Communications Engineering and RF-Systems, Johannes Kepler University, 4040 Linz, Austria; 2Institute for Communications Engineering and RF-Systems, Johannes Kepler University, 4040 Linz, Austria

**Keywords:** physical tamper attack, OFDM, CSI, deep learning, anomaly detection

## Abstract

This paper proposes two deep-learning (DL)-based approaches to a physical tamper attack detection problem in orthogonal frequency division multiplexing (OFDM) systems with multiple receiver antennas based on channel state information (CSI) estimates. The physical tamper attack is considered as the unwanted change of antenna orientation at the transmitter or receiver. Approaching the tamper attack scenario as a semi-supervised anomaly detection problem, the algorithms are trained solely based on tamper-attack-free measurements, while operating in general scenarios that may include physical tamper attacks. Two major challenges in the algorithm design are environmental changes, e.g., moving persons, that are not due to an attack and evaluating the trade-off between detection performance and complexity. Our experimental results from two different environments, comprising an office and a hall, show the proper detection performances of the proposed methods with different complexity levels. The optimal proposed method achieves a 93.32% true positive rate and a 10% false positive rate with a suitable level of complexity.

## 1. Introduction

Wireless networks in critical infrastructures require a high level of security. Therefore, different network security threats need to be considered in such applications. Among them, physical tampering with a device is one that is missing in many applications. As discussed by [1], a possible physical tamper attack is the altering of the orientation of surveillance cameras that monitor a critical infrastructure. Radio-frequency (RF) fingerprint-based localization systems using physical (PHY) layer measurements (e.g., received signal strength indication (RSSI) [2], channel impulse response (CIR) [3], channel state information (CSI) [4], etc.) comprise another example in which physical tampering can significantly distort the system function (e.g., by changing antenna characteristics). According to the European Union Agency for Cybersecurity (ENISA) [5], physical tamper attacks within IoT applications are one of the main threats faced by healthcare organizations as well. An assumption in all these cases is that the transceivers must not be tampered for the system to work correctly. Thus, the functionality of the systems is destroyed with high probability, if the transceivers are tampered. In order to recognize such attacks, a physical tamper attack detection mechanism is required.

To address this issue, radio channel characteristics, which are observed by measurements [1,3,6,7], can help us to detect the tamper attack. Such measurements are the CIR [3], CSI [1,6,7], and received packet features [8,9]. However, the characteristics are not solely influenced by the tamper attack, but also by regular environmental changes. As one of the first works, the proposed CIR method in [3] was only tested in environments with few dynamic elements and, thus, experienced high misdetection rates in dynamic environments. Reference [7] also investigated the feasibility of using a commercial off-the-shelf (COTS) Wi-Fi device as the physical tamper detector based on CSI collection in an almost static environment. To achieve resilience to regular environmental changes, Reference [1] proposed to increase space diversity by using multiple antennas at the receiver. However, others proposed to make use of machine learning (ML) approaches to tackle this issue. Reference [6] showed that a semi-supervised deep learning (DL) algorithm with a postprocessing unit can extract the characteristics of environments and outperforms the approach by [1]. References [8,9] also showed that their ML approaches for the detection of removal/addition of sensors within IoT applications perform with high accuracy in a dynamic environment. The summary of the aforementioned works can be seen in Table 1.

To perform physical tamper attack detection, three main strategies have been applied in the literature. These are: (i) distance computation between previous measurements and new measurements either with hypothesis testing for CIR values in [3] or with direct threshold detection for the CSI in [1]; (ii) distance computation to a lower-dimensional signal representation obtained in a DL framework in [6]; and (iii) a direct detection using the ML algorithms in [7,8,9].

The most recent methods, namely (ii) and (iii), outperform previous methods in terms of the attack detection accuracy. However, it is a cumbersome task to directly compare the aforementioned methods due to their different communication systems. Therefore, since currently, orthogonal frequency division multiplexing (OFDM) is one of the most common transmission technologies [10], we followed [1,6,7] and based our proposed physical tamper attack detection methods on the estimated CSI in an OFDM-based wireless system.

To detect physical tamper attacks in an OFDM-based system, Reference [1] proposed to use multiple antennas at the receiver and calculate the distances between tamper-free CSI in the offline and online phases. This approach was based on the assumption that environmental variations will not affect all CSI received from different antennas at the receiver. In [6], a mixed DL approach, including a deep convolutional autoencoder (DCAE) and a postprocessing, was applied. The DCAE tried to reconstruct the measured CSI with lower-dimensional features. The reconstructed version was then compared to the measurement and used (after robust postprocessing) for attack detection. The disadvantages of [6] were the time delay introduced by the necessary postprocessing unit, no possibility for using multiple CSI estimates at the receiver(s), and the high number of parameters that have to be adjusted manually. In [7], a fully DL approach (cf. [6]) was applied, where a deep neural network with two hidden layers was used. The network used the CSI as the input and output the probability of a tamper attack at predetermined reference positions. While the computational complexity of the method was low, it aimed to identify 1 of N authorized positions, which is a different problem compared to the problem described in [1,6]. Therefore, the method in [7] could not be applied to the problem and was not compared with the proposed methods. (In this work, the performance comparison was made with [1,6] in the Experimental Section).

As almost all modern wireless communication systems support multiple-input multiple-output (MIMO), we were motivated to extend the proposed methods in [6] for the case that multiple CSI estimates at the receiver(s) are available (multi-CSI estimates can be available from either a single receiver with multiple antennas or multiple receivers, each with a single antenna). In this work, we thus expanded the framework of [6] to multi-CSI estimates at the receiver(s). Moreover, the drawback of [6] motivated us to propose using a fully DL approach to detect the physical tamper attack. As shown in [6], a DL approach for dimensionality reduction followed by a postprocessing unit detects the physical tamper attack with a high time delay due to using the postprocessing unit. To solve that, we proposed to use a fully DL approach to simultaneously reduce the dimensionality of the input data along with the anomaly detection task. As stated in [11], support vector data description (SVDD) [12] is one of the popular approaches for anomaly detection. We show that a well-tuned version of the proposed method in [13], namely the Deep SVDD, can be used for physical tamper attack detection. In order to fairly evaluate the physical tamper attack detection methods, both the detection performance and the efficiency (e.g., the time delay due to computational complexity in [6]) were taken into account, which was neglected in previous works. The proposed methods offer different characteristics in detection performance, time complexity, and database space budget.

In summary, this paper extends the aforementioned works in two ways: (i) simultaneous with the detection performance analysis, we evaluate efficiency by means of time delay and database space budget; (ii) centralized or decentralized detectors are considered in the design of the proposed methods. In detail, the contributions of the paper are as follows:Extending the framework for physical tamper attack detection presented in [6] for the case that multiple CSI estimates at the receiver (s) are available: We suggest two distinct approaches, i.e., centralized and decentralized processing. We show that centralized processing has better detection performance and requires lower database space, while having higher time complexity.Proposing the Deep SVDD framework to overcome complexity and latency limitations: We apply Deep SVDD to the physical tamper attack detection problem and show that it has significantly lower complexity compared to the DCAE approach, while having only slightly decreased detection performance.Complexity analysis: We characterize the algorithmic complexity by the number of mathematical operations and required database space to compare all investigated methods. We show that there is a trade-off between detection performance and complexity in the proposed methods.We evaluate all methods on experimental data from a measurement campaign in a university building.

The rest of this paper is organized as follows: Section 2 introduces the tamper attack detection framework. The tamper attack detection methods will be presented in Section 3. The experimental results are discussed in Section 4. Finally, Section 5 concludes the paper.

## 2. Tamper Attack Detection Framework

### 2.1. Problem Statement

As illustrated in Figure 1, considering an OFDM system with one transmitter and *R* antennas in the receiver(s), the transmitter sends regular messages containing pre-defined preambles. Based on the preamble of the *i*th packet, the CSI estimate from the *r*th antenna ${\hat{H}}_{r}^{i}\in C^{1\times S}$ with *S* being the number of subcarriers is obtained. Meanwhile, an attacker intends to physically manipulate the transmitter by relocating and/or reorienting the transmitter. This physical tamper attack has to be detected using the magnitude of CSI values $|{\hat{H}}_{r}^{i}|$, r=1,…,R. (According to [14], we took only the magnitude values as the input to our algorithm and neglected the phase.) Undoubtedly, several challenges have to be addressed to solve this problem.

The transmitted signal is altered by the mobile radio channel. The propagation of the electromagnetic waves from transmitter to receiver(s), as determined by the surrounding environment, determines the mobile radio channel. Any change in the environment also changes the mobile radio channel. As a consequence, the main challenge in detecting a physical tamper attack based on the CSI is to distinguish between changes induced by the physical tamper attack and changes caused by modifications in the surrounding such as people passing by.

Other challenges are time and space constraints during the test phase in applications in which fast system reaction within a restricted database space budget is required. Therefore, time and space complexities have to be taken into account in the evaluation of the proposed methods.

### 2.2. Detection Framework

We formulated the problem of detecting the physical tamper attack as a data-driven semi-supervised anomaly detection problem. Semi-supervised anomaly detection, according to [15], refers to the problem of finding patterns in data that do not correspond to the expected behavior, i.e., finding if the latest measured $|{\hat{H}}_{r}^{i}|$ relates to the attack-free training or not.

The proposed algorithms consist of two phases, i.e., training (offline) and testing (online). According to the structure of the receiver(s), the algorithms are applied on either a centralized or decentralized unit. Collections of tamper-free CSI estimates based on different time and environmental conditions are used for training during the offline phase.

Data are collected in the offline phase for $N_{\mathrm{Off}}^{✩}$ packets, $H_{\mathrm{Off},r}$, and in the online phase for $N_{\mathrm{On}}$ packets, $H_{\mathrm{On},r}$ as below: (1)$$\begin{array}{cc} \; & \forall r\in \{1,\cdots ,R\}:\\ & H_{\mathrm{Off},r}≜\left[\right|{\hat{H}}_{\mathrm{Off},r}^{1}\left|,\right|{\hat{H}}_{\mathrm{Off},r}^{2}\left|,\cdots ,\right|{\hat{H}}_{\mathrm{Off},r}^{N_{\mathrm{Off}}^{✩}}{\left|\right]}^{T}\in R^{N_{\mathrm{Off}}^{✩}\times S}\\ \; & H_{\mathrm{On},r}≜\left[\right|{\hat{H}}_{\mathrm{On},r}^{1}\left|,\right|{\hat{H}}_{\mathrm{On},r}^{2}\left|,\cdots ,\right|{\hat{H}}_{\mathrm{On},r}^{N_{\mathrm{On}}}{\left|\right]}^{T}\in R^{N_{\mathrm{On}}\times S}\;. \end{array}$$

For training, the total number of training samples is split into batches with batch size $N_{\mathrm{Off}}$, such that $N_{\mathrm{Off}}^{✩}$ is an integer multiple of $N_{\mathrm{Off}}$.

The proposed detection framework is illustrated in Figure 2 in which the CSI is estimated from multiple antennas. Based on the structure of the systems, multiple receivers, each with one antenna or a receiver with multiple antennas, we propose the following detection methods: (i) decentralized processing and the combination of anomaly scores; (ii) centralized processing with multi-channel extension of the detection algorithms. The blocks in Figure 2 termed “deep” represent either mixed or fully deep approaches.

## 3. Tamper Attack Detection Methods

In this section, we start with the reformulation of the relevant work [1] in Section 3.1 for the sake of clarity. Afterwards, an extended framework of [6] for multiple antennas at the receiver(s) is proposed in Section 3.2. Finally, a fully DL approach is proposed in Section 3.3 to detect the tamper attack. The proposed methods are compared with respect to detection performance, time complexity, and required database space in the subsequent sections.

### 3.1. Conventional Threshold Detection

The algorithm presented in [1] is a simple and straightforward approach to detect tamper attacks. Distance values $D_{i,j,r}$ between $N_{\mathrm{On}}$ successive CSI estimates in the online phase and $N_{\mathrm{Off}}^{✩}$ recorded values from the offline phase for all *R* receivers are computed as
(2)$$\begin{array}{c} D_{i,j,r}≜Distance(|{\hat{H}}_{\mathrm{Off},r}^{i}\left|,\right|{\hat{H}}_{\mathrm{On},r}^{j}\left|\right) \end{array}$$
where $i=1,\cdots ,N_{\mathrm{Off}}^{✩}$, $j=1,\cdots ,N_{\mathrm{On}}$, r=1,⋯,R and then compared to a threshold (Different threshold selection methods are available in the literature. However, threshold selection methods are out of the scope of this work.) value according to
(3)$$\frac{1}{N_{\mathrm{Off}}^{✩}N_{\mathrm{On}}\;R}\sum_{i=1}^{N_{\mathrm{Off}}^{✩}}\sum_{j=1}^{N_{\mathrm{On}}}\sum_{r=1}^{R}D_{i,j,r}\overset{Tamper\;Free}{\underset{Tampering}{≶}}Threshold\;.$$
As in [1], Distance in (Equation 2) refers to a distance metric (e.g., normalized Euclidean distance). The distance metric quantifies the distance between two CSI vectors. Afterwards, in (Equation 3), the mean value of the distance over all data is considered to make the decision. This makes the method quite complex since (Equation 2) has to be computed for the entire training data set. In this work, Threshold was set based on a given performance detection (i.e., false positive rate (FPR)). In Section 4.6.1, we compare the proposed method at FPR = 10%).

To distinguish between conventional environmental changes and a tamper attack, Reference [1] relies on diversity introduced by multiple receiver antennas. This follows the assumption that, e.g., a moving person will impact the CSI only on a subset of the receivers, while the tamper attack impacts the CSI at all receivers. Consequently, the attack detection performance is poor for a single-receiver system. In this work, this approach is referred to as **Threshold Detection**.

### 3.2. Mixed Deep Approaches: DCAE with PDF Estimator

As an extension of the work in [6], during the offline phase, DCAE_r_ (see Figure 3) learns a low-dimensional representation of the tamper-free CSI, $|H_{\mathrm{Off},r}^{i}|$, for $i=1,\cdots ,N_{\mathrm{Off}}^{✩}$ and r=1,⋯,R. From this representation, the CSI is reconstructed (referred to as $|{\hat{H}}_{r}^{i}{|}_{\mathrm{rec}}$) and the reconstruction error $e_{r}^{i}$ is computed. After training the network, the Euclidean norm of the reconstruction error is used as the anomaly score $a_{r}$ as: (4)$$\begin{array}{c} a_{r}=\left[\right\|e_{r}^{1}{\|}_{2},\|e_{r}^{2}{\|}_{2},\cdots ,\|e_{r}^{N_{\mathrm{Off}}^{✩}}{{\|}_{2}]}^{T}\epsilon \;R^{N_{\mathrm{Off}}^{✩}}\;. \end{array}$$

The overall scheme for the *r*-th receiver is depicted in Figure 3 and Figure 4, for the offline and online phases, respectively. For training, the offline CSI estimates in the *r*-th receiver (denoted by $H_{\mathrm{Off},r}$ in Figure 3) are tamper-free under different environmental conditions including the movement of people and static environments at different times. Then, the trained DCAE_r_ (denoted as ${\mathrm{DCAE}}_{r}^{*}$ (the asterisk superscripts indicate the trained versions of DCAE)) is used for calculating the anomaly score $a_{r}$ of the *r*-th receiver.

To increase the robustness of physical tamper attack detection, a pdf estimation is applied to the reconstruction error to capture the statistics of the variation in the tamper-free scenario. A straightforward pdf estimation approach is non-parametric kernel density estimation [16]. Let $a_{r}\overset{\Delta }{=}\left(a_{r}^{\left(1\right)},a_{r}^{\left(2\right)},\cdots ,a_{r}^{\left(N\right)}\right)$ be independent and identically distributed samples drawn from some univariate distribution with an unknown density f[a] at any given point *a*. Thus, the kernel density estimator is:(5)$$\hat{f}\left[a\right]=\frac{1}{Nh}\sum_{i=1}^{N}K\left[\frac{a-a_{r}^{\left(i\right)}}{h}\right],$$
where *h* is a smoothing parameter referred to as the bandwidth and K[.] is the kernel function. In this work, *h* is calculated using Silverman’s rule of thumb [17] as:(6)$$h=0.9\min\left(\hat{\sigma },\frac{IQR}{1.34}\right)N^{-\frac{1}{5}},$$
where $\hat{\sigma }$ is the standard deviation of the samples, IQR is the interquartile range, and *N* is the sample size. Herein, we make use of the Gaussian kernel function to estimate the pdfs.

To evaluate the anomaly score for *R* receiver antennas, a decentralized and a centralized approach are considered. Obviously, data processing at each receiver is preferred in cases in which multi-CSI estimates are available from multiple receivers, whereas centralized data processing is preferred in cases in which multi-CSI estimates are available from a single receiver with multiple antennas.

#### 3.2.1. Decentralized Processing with Multiple DCAEs

If decentralized processing is chosen, a single DCAE is applied at each receiver. As shown in the subsequent sections, the size of the input data for this approach is [$N_{\mathrm{Off}}$, 1, *S*]. The weights of ${\mathrm{DCAE}}_{r}^{*}$ and its pdf estimation ${\hat{f}}_{Y_{r,\mathrm{Off}}}\left[a\right]$ are stored in the database for further actions in the online phase.

For each receiver, we used the overlapping index approach [18] to measure the distance of the online-phase pdf ${\hat{f}}_{Y_{r,\mathrm{On}}}\left[a\right]$ to the offline-phase pdf (see Figure 4). The overlapping index distance measure for the *r*-th receiver, $\eta_{r}:\;R^{n}\;\times \;R^{n}\to \;\left[0,1\right]$, is defined as:(7)$$\begin{array}{cc} \; & \forall r\in \{1,\cdots ,R\}:\\ \; & \eta_{r}\left({\hat{f}}_{Y_{r,\mathrm{Off}}}\left[a\right],{\hat{f}}_{Y_{r,\mathrm{On}}}\left[a\right]\right)=\sum_{i=1}^{n}\min\left\{{\hat{f}}_{Y_{r,\mathrm{Off}}}\left[a_{i}\right],{\hat{f}}_{Y_{r,\mathrm{On}}}\left[a_{i}\right]\right\}\; \end{array}$$
where ${\hat{f}}_{Y_{r,\mathrm{Off}}}\left[a\right]$ and ${\hat{f}}_{Y_{r,\mathrm{On}}}\left[a\right]$ are the pdf approximations of the anomaly score in the offline and online phase of the *r*-th receiver, respectively. The anomaly score is averaged across the *R* receivers and compared to a threshold to decide about an attack, i.e.,
(8)$$\frac{1}{R}\sum_{r=1}^{R}\eta_{r}\overset{Tamper\;Free}{\underset{Tampering}{≶}}Threshold\;.$$

In this work, two different approaches were considered: (i) Using DCAE without pdf estimator unit: The approach is the same as in Figure 3 and Figure 4, but the pdf estimator unit is exchanged with a unit that calculates the mean value of its inputs. This approach is referred to as **DCAE-D** (**D** for decentralized). (ii) Using DCAE with the pdf estimator unit: The approach is depicted in Figure 3 and Figure 4, which is referred to as **DCAE-DP** (**DP** for decentralized and postprocessing).

#### 3.2.2. Centralized Processing with a Single Multi-Channel DCAE

Here, we propose to use a single DCAE, which is capable of learning the combination of input data from all *R* receiver antennas. As shown in the subsequent sections, the size of the input for this structure is [$N_{\mathrm{Off}}$, *R*, *S*]. This approach is also enhanced in terms of detection performance by utilizing a pdf estimator. In the offline phase (see Figure 5), the weights of DCAE^*^ and its pdf estimation of the anomaly scores are stored in the database.

In the online phase, DCAE^*^ is used to calculate the anomaly score of newly received CSI estimates. Afterward, its pdf approximation is compared with the pdf stored in the database (see Figure 6). The overlapping index distance measure for this approach, $\eta :\;R^{n}\;\times \;R^{n}\to \;\left[0,1\right]$, is defined as:(9)$$\eta \left({\hat{f}}_{Y_{\mathrm{Off}}}\left[a\right],{\hat{f}}_{Y_{\mathrm{On}}}\left[a\right]\right)=\sum_{i=1}^{n}\min\left\{{\hat{f}}_{Y_{\mathrm{Off}}}\left[a_{i}\right],{\hat{f}}_{Y_{\mathrm{On}}}\left[a_{i}\right]\right\}\;$$
where ${\hat{f}}_{Y_{\mathrm{Off}}}\left[a\right]$ and ${\hat{f}}_{Y_{\mathrm{On}}}\left[a\right]$ are the pdf approximations of the anomaly score in the offline and online phase, respectively. The anomaly score is compared to a threshold to decide about an attack, i.e.,
(10)$$\eta \overset{Tamper\;Free}{\underset{Tampering}{≶}}Threshold\;.$$
As will be shown in Section 4.6.2, by utilizing one DCAE instead of multiple DCAEs, the computational complexity is significantly reduced.

In this work, the approach that utilizes the multi-channel DCAE without a pdf estimator is considered as **DCAE-C** (**C** for centralized) and the one with a pdf estimator is regarded as **DCAE-CP** (**CP** for centralized and postprocessing).

### 3.3. A Fully Deep Approach: Deep SVDD

In this subsection, we exploit Deep SVDD, a novel method proposed by [13], for solving the physical tamper attack detection problem. Deep SVDD is a method that is usually utilized for one-class classification problems in image processing applications. It is a fully deep approach for anomaly detection, which maps the input space into an output space (i.e., a hypersphere of minimum volume) with a neural network (see Figure 7). Feature representations of the data, as well as the one-class classification objective are learned by the neural network. Similar to the previous methods, two different approaches were considered for multi-receiver operation: either using a neural network for each receiver or using a neural network that accepts multiple inputs. We first reformulated the notation of the method of Deep SVDD [13] to be suitable for the problem at hand. Afterwards, we discuss the details of each approach.

Let ϕ(.;W):χ→F be a neural network with L∈N hidden layers and a set of weights $W=\{W^{1},\cdots ,W^{L}\}$, where $W^{l}$ are the weights of layer l∈{1,⋯,L} for some input space $\chi \subseteq R^{S}$ and output space $F\subseteq R^{P}$. The feature representation of $|\hat{H}|\in \chi$ is $\phi (|\hat{H}|;W)$, i.e., the network ϕ with the weights W. Then, the objective is to learn the network weights W while minimizing the volume of a data-enclosing hypersphere in output space F, which is defined by radius R and center c. Given the training data $D_{N_{\mathrm{Off}}^{✩}}=\left\{\right|\hat{H}{|}_{1},\cdots ,|\hat{H}{|}_{N_{\mathrm{Off}}^{✩}}\}$ on χ, the objective function is defined as: (11)$$\min_{W}\frac{1}{N_{\mathrm{Off}}^{✩}}\sum_{i=1}^{N_{\mathrm{Off}}^{✩}}\left\|\phi (\right|\hat{H}{|}_{i}{;W)-c\|}^{2}+\frac{\lambda }{2}\sum_{l=1}^{L}{\left\|W^{l}\right\|}_{F}^{2}\;.$$
The first term in (Equation 11) considers the distance of each network representation $\phi (|\hat{H}{|}_{i};W)$ to c∈F. The second term is a network weight decay regularizer with hyperparameter λ to reduce overfitting of the DL model, where ${\left\|.\right\|}_{F}$ refers to the Frobenius norm. According to [13], c∈F is any fixed hypersphere center.

In the offline phase, the Deep SVDD network is trained using tamper-free CSI estimates (i.e., training data $D_{N_{\mathrm{Off}}^{✩}}$) from different environmental conditions, such as static environments and the movement of people at different times with a batch size of $N_{\mathrm{Off}}$. After training the Deep SVDD network, the weights of the trained network $W^{*}$ and the hypersphere in output space F, which is defined by radius $R^{*}$ and center $c^{*}$, are stored in the database for the following use in the online phase.

In the online phase, the anomaly score of each new estimate, $|{\hat{H}}_{\mathrm{On}}|\in \chi$, is calculated by the distance of $|{\hat{H}}_{\mathrm{On}}|$ to the center of the hypersphere, i.e.,
(12)$$s(|{\hat{H}}_{\mathrm{On}}\left|\right)=\left\|\phi (\right|{\hat{H}}_{\mathrm{On}}|;W^{*}{)-c^{*}\|}^{2}\;.$$

As stated, two different approaches are considered for multi-receiver operation. Therefore, the decision is made based on the following approaches in the online phase.

#### 3.3.1. Multiple Neural Networks in Decentralized Mode

For using *R* SVDD-based detectors in decentralized mode, each is trained according to the objective function: (13)$$\min_{W_{r}}\frac{1}{N_{\mathrm{Off}}^{✩}}\sum_{i=1}^{N_{\mathrm{Off}}^{✩}}\|\phi_{r}\left(\right|{\hat{H}}_{r}{|}_{i};W_{r})-c_{r}{\|}^{2}+\frac{\lambda }{2}\sum_{l=1}^{L}{\left\|W_{r}^{l}\right\|}_{F}^{2}\;,$$
for r=1,⋯,R. In the online phase, the attack detection follows the threshold detection:(14)$$\frac{1}{R}\sum_{r=1}^{R}\|\phi_{r}\left(\right|{\hat{H}}_{r}|;W_{r}^{*}{)-c_{r}^{*}\|}^{2}\overset{Tamper\;Free}{\underset{Tampering}{≶}}R^{*}\;,$$
i.e., the average of the distance between the mapped inputs and the stored hypersphere centers $c_{r}^{*}$ is compared with the stored radius $R^{*}$. (In this work, $R^{*}$ was set based on a given performance detection (i.e., FPR)).

#### 3.3.2. Multi-Channel Single Neural Network

In this approach, a single neural network is utilized, which is capable of learning all data from *R* receivers simultaneously. Therefore, its objective function is: (15)$$\min_{W}\frac{1}{N_{\mathrm{Off}}^{✩}R}\sum_{r=1}^{R}\sum_{i=1}^{N_{\mathrm{Off}}^{✩}}\left\|\phi (\right|\hat{H}{|}_{i,r}{;W)-c\|}^{2}+\frac{\lambda }{2}\sum_{l=1}^{L}{\left\|W^{l}\right\|}_{F}^{2}\;.$$
The decision is made based on the following equation:(16)$$\left\|\phi (\right|\hat{H}|;W^{*}{)-c^{*}\|}^{2}\overset{Tamper\;Free}{\underset{Tampering}{≶}}R^{*}\;.$$
It is worth noting that the network weights $W_{r}^{*}$, $R^{*}$ and $c_{r}^{*}$, for r=1,⋯,R, in the decentralized approach (see (Equation 14)), and the network weights $W^{*}$, $R^{*}$, and $c^{*}$ in the centralized approach (see (Equation 16)) are sufficient to characterize the multiple Deep SVDD models and the multi-channel Deep SVDD model, respectively. No further data need to be stored in the database for physical tamper attack detection. In contrast, for the DCAE-based approaches, not only the weights of the trained network(s), but also a representation of the offline anomaly scores (i.e., the trained pdf) and the threshold have to be stored. As a result, it is expected that Deep SVDD has a lower database space requirement, which leads to faster testing and lower time delay in the online phase. Moreover, the pdf estimator in the DCAE-based methods relies on $N_{\mathrm{On}}$ successive CSI estimates in the online phase (i.e., referred to as the online batch). In contrast to the DCAE-based methods, the value of $N_{\mathrm{On}}$ in the tamper attack decision for Deep-SVDD-based methods does not significantly affect the detection performance (shown in Section 4.5.3).

In this work, the decentralized and centralized Deep SVDD approaches are referred to as **SVDD-D** (**D** for decentralized) and **SVDD-C** (**C** for centralized), respectively.

## 4. Experimental Results

The presented methods were evaluated and compared in a setup in which two receivers, each equipped with a single antenna, should detect a tamper attack performed at the transmitter, also equipped with a single antenna. As the tamper attack, we considered a rotation of the transmitter compared to its default orientation. The tamper attack detection is based on CSI estimates acquired with a software-defined radio and the Gnuradio OFDM project [19]. The parameters were selected as described in the following. (Note that the underlying OFDM estimates are identical to those in our previous work in [6]).

### 4.1. OFDM System

The transmitter and receiver nodes were composed of a host computer connected to a USRP X310 equipped with a directional antenna [20]. In the system, data were exchanged among nodes with a frame structure. Each frame consisted of nine data OFDM symbols and three preamble symbols. In the frequency domain, each symbol contained 200 data subcarriers, 48 null subcarriers, and 8 pilot subcarriers for a channel bandwidth of 25 MHz. Each OFDM symbol duration consisted of a 10.24 μs IFFT period followed by a 1.25 μs guard interval. The carrier frequency was set to 2.55 GHz. The channel was estimated based on one preamble symbol with a least-squares approach. The channel estimation (i.e., denoted by $\hat{H}$) was used in the aforementioned physical tamper attack detection methods.

### 4.2. Environment

As seen in Figure 8, we evaluated the tamper attack detection methods in two distinct environments: an office and a hall. The transmitter (indicated by TX) and two receivers (indicated by RX1 and RX2) were placed on the top of shelves with a 230 cm elevation in the office and desks with a 140 cm elevation in the hall environment. To have physical tamper attacks and tamper-free scenarios in the estimates, we took into account eight different antenna orientations, including the tamper-free default orientation and rotations r1, r2, … r7 (cf. Figure 8).

Since discriminating between environmental changes and physical tamper attacks is the key challenge in this attack detection problem, we considered eight scenarios for the tamper-free default orientation (see Table 2). In these eight scenarios, typical situations in an office and a hall were considered in which small-scale (scenarios A and B) and large-scale (scenarios C to H) movements of persons appear, which introduce variations in the CSI. These CSI variations due to the movement of people (i.e., environmental changes) need to be learned for our proposed methods to be able to distinguish them from the CSI changes due to physical tamper attacks.

### 4.3. Parameters of the DL-Based Methods

For all neural networks in this work, we used LeNet-type convolutional neural networks (CNNs). In the following, two structures are described in Figure 9 and Figure 10, which were used for the aforementioned methods.

#### 4.3.1. DCAE

The DCAE-based methods require parameter selection for the DCAE core block (responsible for representation learning) and the postprocessing unit (responsible for anomaly detection). For the latter, the free parameter is the online batch size $N_{\mathrm{On}}$. For the DCAE, each convolutional module is composed of a one-dimensional convolutional layer followed by ELU activation functions (as depicted in Figure 9 with Conv1D), BatchNorm1D, and 2×1 Max-Pooling layers. The transposed convolutional modules are composed of a one-dimensional transposed convolutional layer followed by ELU activation functions (as depicted in Figure 9 with ConvT1D), BatchNorm1D, and 2×1 Interpolate layers. As depicted in Figure 9, the encoder is composed of three convolutional modules with different numbers of filters (eight, four, and one) and a final fully dense layer of 26 units. The mirrored structured is used for the decoder. The colors in Figure 9 show the operations that were applied. Dark red, light red, dark blue, and light blue color tensors illustrate that a convolutional module, BatchNorm1D followed by 2×1 Max-Pooling, BatchNorm1D followed by 2×1 Interpolate layers, and a transposed convolutional module were applied on the previous corresponding tensor, respectively. The black color tensor indicates the compressed representation of the input. The data flow starts from the top left-hand side and ends at the bottom left-hand side.

#### 4.3.2. Deep SVDD

As stated in Section 3.3, representation learning and anomaly detection are simultaneously performed in Deep SVDD. Therefore, the structure of the encoder of DCAE is adopted for the structure of Deep SVDD (cf. Figure 9 and Figure 10). By this approach, the time complexity along with the space complexity of the Deep-SVDD-based methods compared to the DCAE-based methods are reduced. (Note that time and space complexities in the ML context are equivalent to computational time and memory requirements in communications engineering).

### 4.4. How to Train the Networks

As Table 3 shows, for the training, validation, and testing data, different CSI estimate datasets from the office and the hall environment were collected. To observe the genuine performance results of the neural networks, they were trained in the default antenna orientation in scenarios A, C, E, and G. Other scenarios and other antenna orientation were utilized for evaluating and testing.

A variant of stochastic gradient descent (i.e., Adam [21]) was utilized to optimize the weights for each method using backpropagation. We made use of the Keras library [22] and implemented Deep SVDD in Pytorch [23].

With an initial learning rate of 10^−5^ and then 10^−6^, we used a two-phase learning rate schedule (searching and fine-tuning). We trained 20 epochs with the learning rate for searching and 5 epochs with the learning rate for fine-tuning. This was repeated 300 times to obtain precise results. The networks were implemented on TensorFlow with Linux (ubuntu 18.04) running on an 8-core ARM v8.2 64-bit CPU and a 512-core NVIDIA Volta GPU. Table 4 summarizes the parameters used to train the neural networks.

The learning curves of the two neural networks are shown in Figure 11. We defined a training batch size of 200. It is worth noting that **DCAE-DP** and **DCAE-CP** use the same neural networks as **DCAE-D** and **DCAE-C**, respectively. Hence, their learning curves are the same as the learning curves of **DCAE-D** and **DCAE-C**, respectively, and they are not plotted in Figure 11. Moreover, in **DCAE-D** and **SVDD-D**, there are *R* (i.e., two in this work) neural networks. Therefore, there are two learning curves for these methods in Figure 11.

Figure 11 shows that the neural networks were appropriately trained. For an easier interpretation, the quadratic loss functions for Deep SVDD and the other DL methods are not normalized in the figure. Therefore, a shift between the learning curves of the Deep-SVDD-based methods and the DCAE-based methods is observed in Figure 11.

### 4.5. Evaluation Criteria

#### 4.5.1. AUC-ROC

The receiver operating characteristic (ROC) is a graphical representation of classifier performance. At various threshold settings, the ROC curve plots the true positive rate (TPR) versus the FPR. The most important evaluation metric for the performance of any classification model is the area under the curve (AUC) of the ROC. We measured the AUC-ROC of the different methods where an excellent one has an AUC close to one, which means it has an almost perfect capability of separation.

#### 4.5.2. Complexity

The complexity of an algorithm indicates the number of features or terms included in the algorithm to execute its task. Time and space are two main complexity measures of the efficiency of an algorithm.

Time complexity is the number of operations in an algorithm required to complete its task with respect to the input size. It can be experimentally evaluated by measuring the time the algorithm requires to accomplish the task. Space complexity denotes the amount of space used by the algorithm for its task, for various input sizes. In this work, we considered the database space the algorithm requires to accomplish its task as the space complexity. We evaluated the time complexity experimentally by runtime measurements of the algorithms and theoretically by the number of basic operations (multiplications and additions) for the forward path. We evaluated the space complexity of an algorithm by the number of floating point elements that are required. Note that time and space complexity depends on other factors as well, including hardware, operating system, processors, other running programs, etc. Since we established the same conditions for the aforementioned algorithms while analyzing them, we did not consider any of these factors in their evaluation.

#### 4.5.3. Detection Performance Alongside Complexity

An ideal tamper-detection method is achieved when the detection performance is high and the complexity is low. However, there is usually a trade-off between the detection performance and the complexity. To fairly evaluate the tamper attack detection methods, the detection performance along with the complexity should be considered.

### 4.6. Tamper Attack Detection Performance

The aim of this subsection is to compare the performance of the proposed methods in terms of attack detection, time complexity, and space complexity. Furthermore, environmental dependency along with detection performance versus complexity are discussed.

#### 4.6.1. ROC Evaluation

The comparison of the detection performance of the aforementioned methods is shown in Figure 12. In this work, similar to [7], we chose the threshold in Figure 12 such that it maximizes the TPR and minimizes the FPR in almost all methods, resulting in an FPR = 10%, which is indicated by the dashed-dotted line. As can be seen from Figure 12, the DCAE-based approaches have a better tamper attack detection rate compared to the Deep SVDD approaches. This is because they spend more effort on signal processing (representing the learned CSI and the postprocessing), which results in higher time and space complexity. As illustrated in Figure 12, **DCAE-C** and **DCAE-CP** have the best tamper attack detection rates. The Deep SVDD approaches have a better tamper attack detection rate compared to the threshold detection approach, although they have lower space complexity.

#### 4.6.2. Time and Space Complexities

The number of basic operations (multiplications and additions) in the forward path that each method requires is compared in Table 5. Furthermore, the time each method required in the offline and online phases was measured and is shown in Table 6. There is a direct relation between time complexity and test duration for each method (cf. Table 5 and Table 6). The amount of data each method stores in the database is another important factor, at least in some applications. Thus, they are presented in Table 7.

As stated, the drawback of the DCAE-based method proposed in [6] is its time delay. The reason is that the pdf estimations in the online phase are time consuming. This issue is addressed in this paper by using the Deep SVDD approach. According to Table 5 (number of basic operations) and Table 6 (test duration), it is shown that the test duration for **SVDD-D** and **SVDD-C** is significantly shortened compared to **DCAE-D** (31.5%), **DCAE-DP** (72.6%), **DCAE-C** (39.0%), and **DCAE-CP** (62.8%), respectively. Note that there is a slight difference in the training duration between the hall and office environment due to the different sizes of their datasets (cf. Table 3).

#### 4.6.3. Environmental Dependency

We can conclude that physical tamper attack detection in the hall environment is more challenging than in the office environment. As shown in Table 6, the attack detection performance of each method in the hall environment is poorer than the corresponding one in the office environment in terms of the AUC. The reason is that the surrounding environment in the hall is prone to be more time-variant compared to the office environment since the environment is not bounded on two sides. Thus, there are more possibilities of people moving around in the hall than in the office. As we used approximately the same amount of training data in both environments, we could not learn the environmental variation to the same amount as in the office environment. We expect that, with a more extensive training, the tamper detection performance can reach similar values as in the office environment.

#### 4.6.4. Detection Performance vs. Complexity

According to Table 6, using **DCAE-CP** not only has superior detection performance compared to the other methods, but shows also relatively fast training and testing phases. In general, there is a trade-off among time complexity, space complexity, and detection performance. As shown in Table 5 and Table 7, **SVDD-D** and **SVDD-C** show the smallest time and space complexity, but as expected, their detection performance is not as high as the others. In contrast, **DCAE-D** and **DCAE-DP** have a larger space complexity than **DCAE-CP**, but achieve a lower detection performance. All in all, the aforementioned factors have to be considered simultaneously to select the optimal method.

It is worth noting that, although the detection performances of **SVDD-D** and **SVDD-C** are acceptable, but not as high as the performances of the other DL methods, their time and space complexities are very low and, thus, might be suitable for certain applications.

As brought up in [6], although the time delay will be longer by increasing the online batch size for **DCAE-DP**, the detection performance will be enhanced. Therefore, we investigated the impact of the online batch size on the detection performance. Figure 13 plots the AUC-ROC over different values of $N_{\mathrm{On}}$. Since $N_{\mathrm{On}}$ significantly influences the performance of the pdf estimators, only the detection performances of **DCAE-DP** and **DCAE-CP** change over different values of $N_{\mathrm{On}}$. As expected, by increasing $N_{\mathrm{On}}$, the AUC-ROC improves. Note that the value of $N_{\mathrm{On}}$ in the depicted range does not affect the performance of **DCAE-CP** in the office environment because the method can learn environmental variation very well. Obviously, the detection performances of the DCAE-based methods with postprocessing will be affected if the value of $N_{\mathrm{On}}$ is chosen too small. According to Figure 13 and the relation between $N_{\mathrm{On}}$ and time complexity, the optimal value of $N_{\mathrm{On}}$ for **DCAE-CP** in the hall environment is 200.

### 4.7. Further Discussion

The structures of **DCAE-C** and **DCAE-CP** are similar to those of **DCAE-D** and **DCAE-DP**, respectively. Instead of using *R* neural networks in **DCAE-D** and **DCAE-DP**, a single multi-channel input neural network is utilized in **DCAE-C** and **DCAE-CP**. According to Table 7, the space complexity of each centralized method is almost *R*-times less than the corresponding decentralized method. In general, the centralized methods outperform the decentralized method in terms of the AUC and space complexity. This is because the centralized methods consider the CSI from all receivers together in the offline phase. However, this is not the case for the SVDD-based method in the hall environment. The reason is the low complexity of the method and the high variation of its signal due to the environment structure. We expect that, with a more complex neural network in the structure of the SVDD-based method, the tamper attack detection performance can achieve similar values as in the office environment.

As stated in [6], DCAE is a method for representation learning. Physical tamper attack detection with DCAE-based methods is performed by the postprocessing units. With Deep SVDD, representation learning and physical tamper attack detection are performed simultaneously. Therefore, increasing the complexity of the neural networks used for Deep SVDD leads to improved detection performance. As in our work, the goal was to minimize complexity, we restricted the number of hidden layers to four in the neural networks used for Deep SVDD. A larger number of hidden layers could have improved the detection performance, but would have increased thecomplexity.

From Table 5 and Table 6, we found that **Threshold Detection** performs much faster than the other methods; however, its tamper attack detection performance is rather low. It is worth mentioning that there is a linear relation between the number of collected packets in the offline phase ($N_{\mathrm{Off}}^{✩}$) and the time complexity for **Threshold Detection** (see Table 5). However, $N_{\mathrm{Off}}^{✩}$ does not affect the time complexity for the DL methods. This is a significant advantage of DL methods over non-DL methods.

## 5. Conclusions

In this paper, we proposed and evaluated two DL-based approaches for detecting a physical tamper attack using CSI in an OFDM-based wireless communication system with multiple receiver antennas with respect to detection performance, time, and space complexities. The two main challenges of this problem were to distinguish between antenna orientation changes and communication environment changes and to achieve high detection performance along with a low level of complexity. To achieve a robust attack detector based on different levels of complexity, we used DCAE and Deep SVDD neural networks in centralized and decentralized structures. With our experiment, we concluded that there is a trade-off between detection performance and complexity in the proposed methods. It was shown that the DCAE-based methods outperform the SVDD-based methods in terms of detection, while the SVDD-based methods have almost two-times lower time and space complexities.

## Figures and Tables

**Figure 1 sensors-22-06547-f001:**
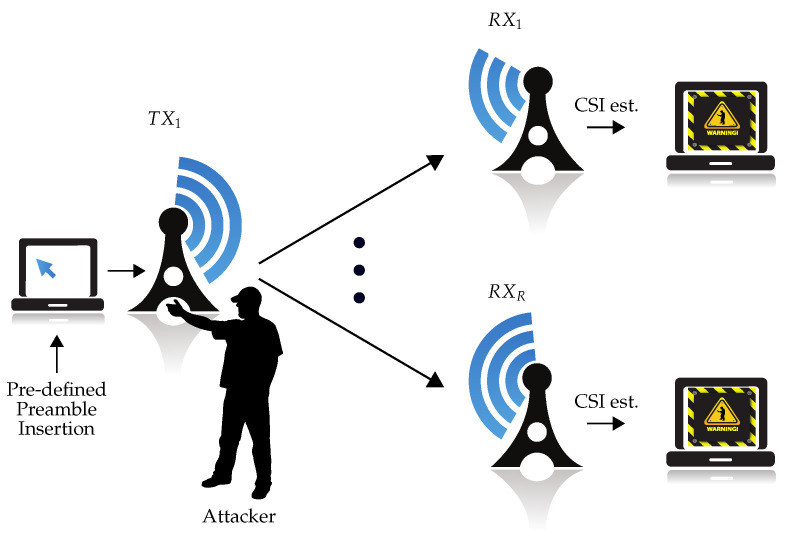
Problem statement.

**Figure 2 sensors-22-06547-f002:**
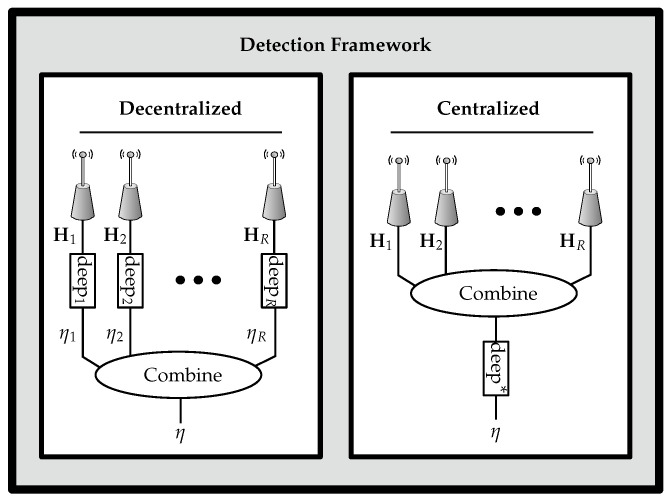
Proposed detection framework.

**Figure 3 sensors-22-06547-f003:**
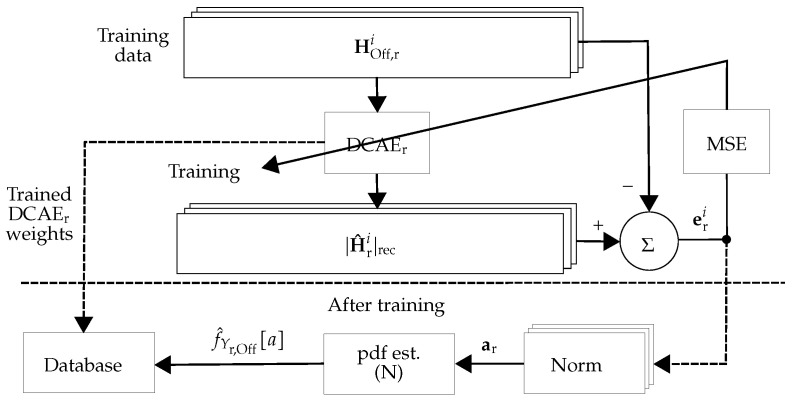
Schematic view of the offline phase of decentralized processing: In the offline phase of the *r*-th receiver, DCAE_r_ is trained. Then, the pdf approximation of the anomaly score (${\hat{f}}_{Y_{r,\mathrm{Off}}}\left[a\right]$) and the weights of the trained network (${\mathrm{DCAE}}_{r}^{*}$) are stored in the database.

**Figure 4 sensors-22-06547-f004:**
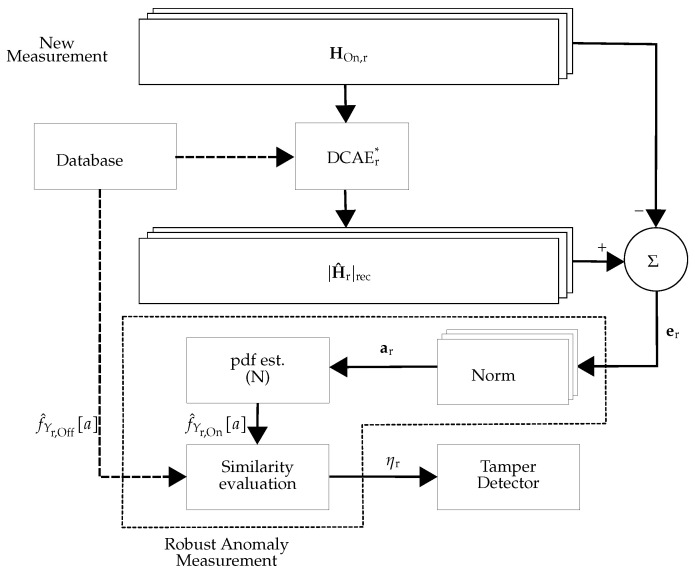
Schematic view of the online phase of decentralized processing: In the online phase of the *r*-th receiver, ${\mathrm{DCAE}}_{r}^{*}$ is used to calculate the anomaly score and the pdf approximation of the anomaly score (${\hat{f}}_{Y_{r,\mathrm{On}}}\left[a\right]$).

**Figure 5 sensors-22-06547-f005:**
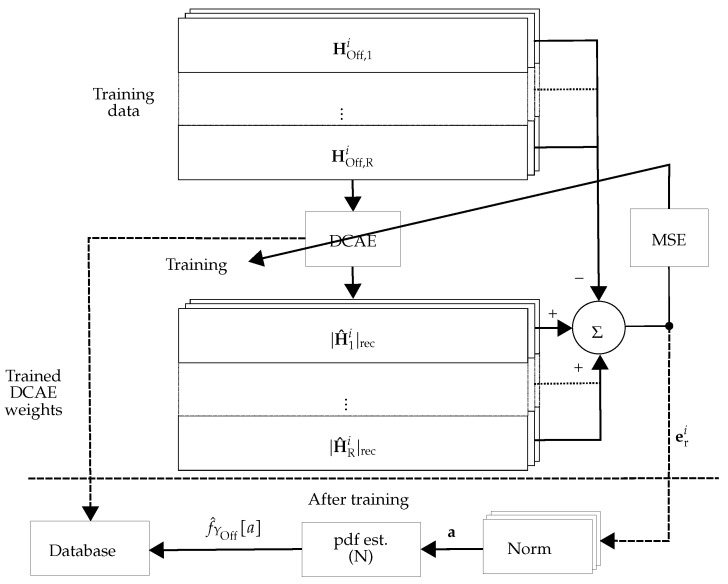
Schematic view of the offline phase of centralized processing: In the offline phase, DCAE is trained. Then, the pdf approximation of the anomaly score (${\hat{f}}_{Y_{\mathrm{Off}}}\left[a\right]$) and the weights of the trained network (DCAE^*^) are stored in the database.

**Figure 6 sensors-22-06547-f006:**
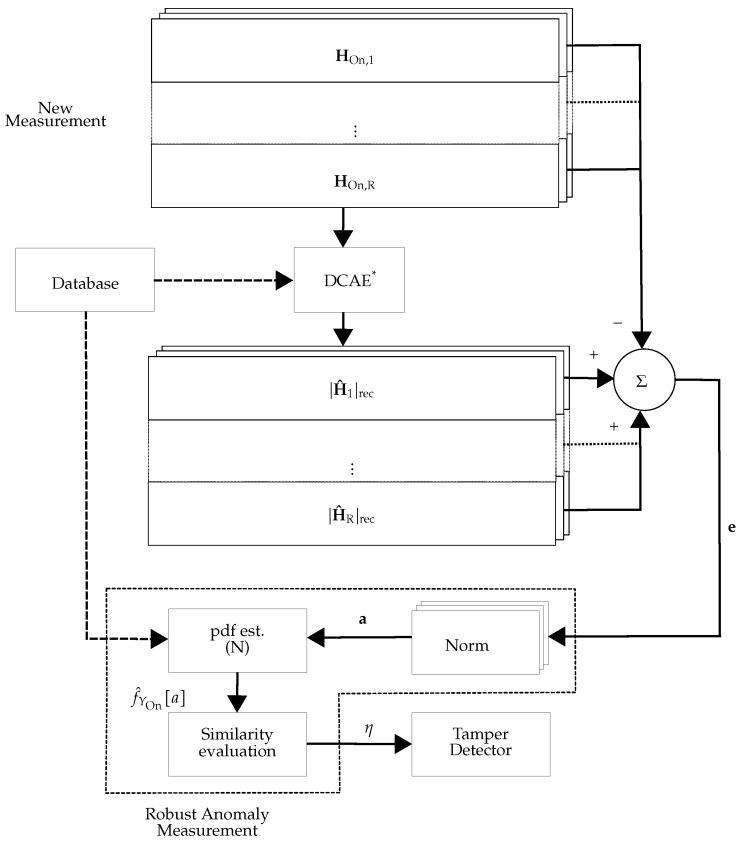
Schematic view of the online phase of centralized processing: In the online phase, DCAE^*^ is used to calculate its anomaly score and the pdf approximation of the anomaly score (${\hat{f}}_{Y_{\mathrm{On}}}\left[a\right]$).

**Figure 7 sensors-22-06547-f007:**
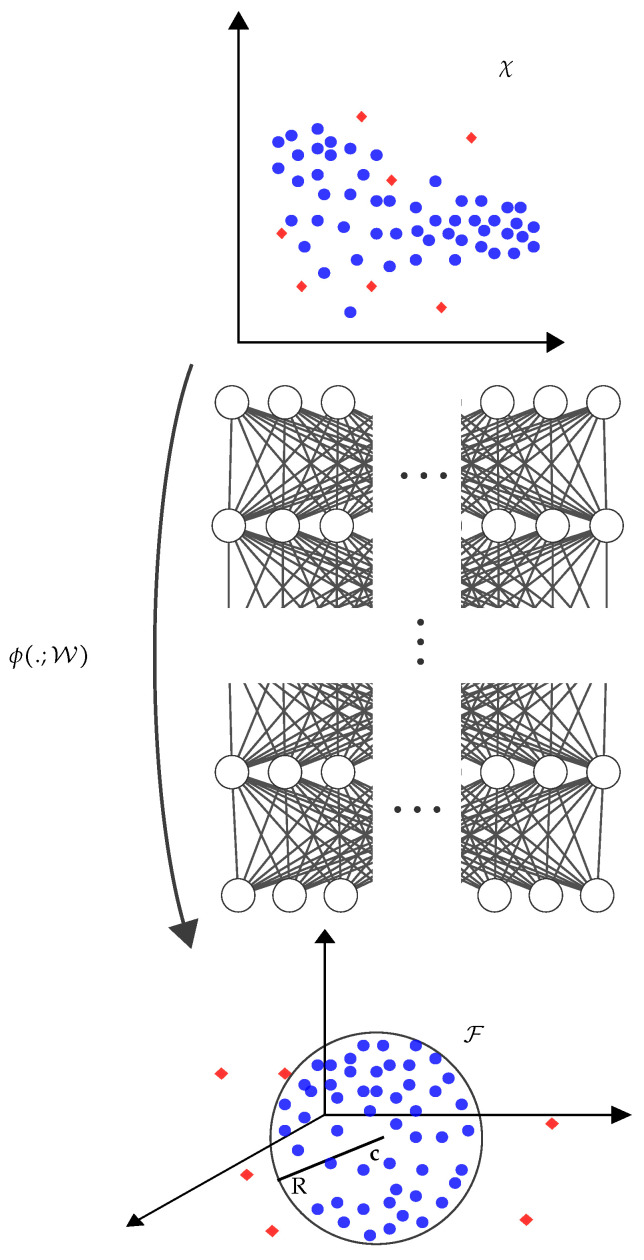
Deep SVDD maps most of the input data into a hypersphere specified by center c and radius R of a minimum volume using a neural network ϕ(.;W) with weights W.

**Figure 8 sensors-22-06547-f008:**
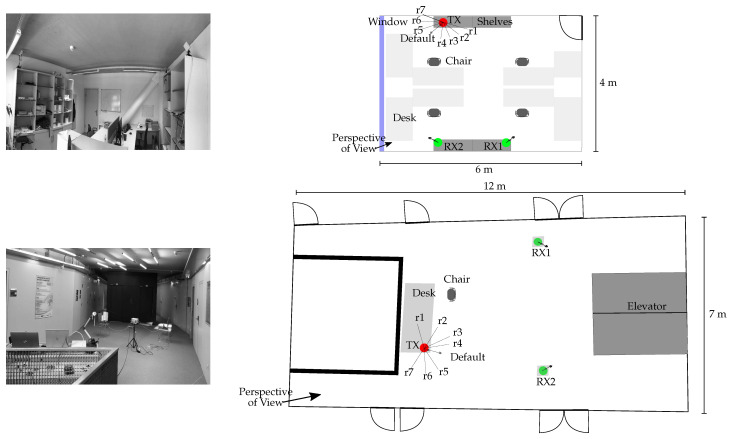
Measurement environments: (top) office (4 m × 6 m) and (bottom) hall (7 m × 12 m), depicted with photos and the layout. The transmitter (indicated by TX) and two receivers (indicated by RX1 and RX2) are denoted. Orientations r1, r2, …, r7 are considered as physical tamper attacks.

**Figure 9 sensors-22-06547-f009:**
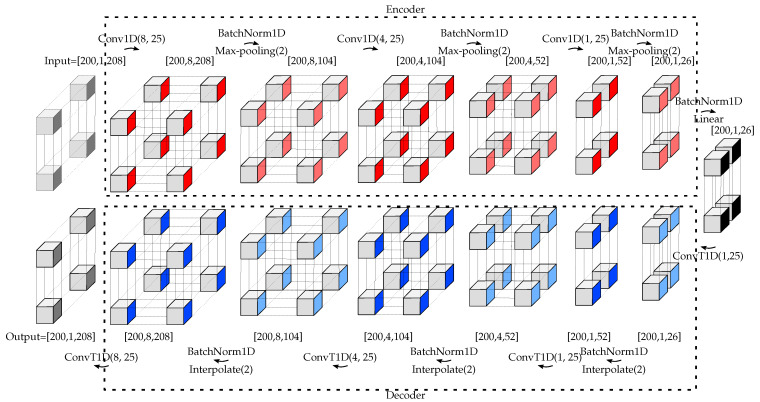
Structure of DCAE. The size of each tensor is depicted in the figure. This structure is used for the proposed decentralized approaches. In the case of centralized approaches, only the size of the input is changed (in the second dimension, 1 is replaced by *R*).

**Figure 10 sensors-22-06547-f010:**
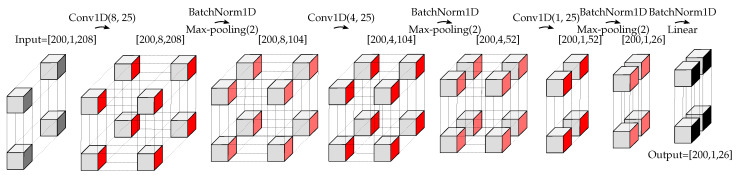
Structure of Deep SVDD. The size of each tensor is depicted in the figure. This structure is used for the proposed decentralized approaches. In the case of centralized approaches, only the size of the input is changed (in the second dimension, 1 is replaced by *R*).

**Figure 11 sensors-22-06547-f011:**
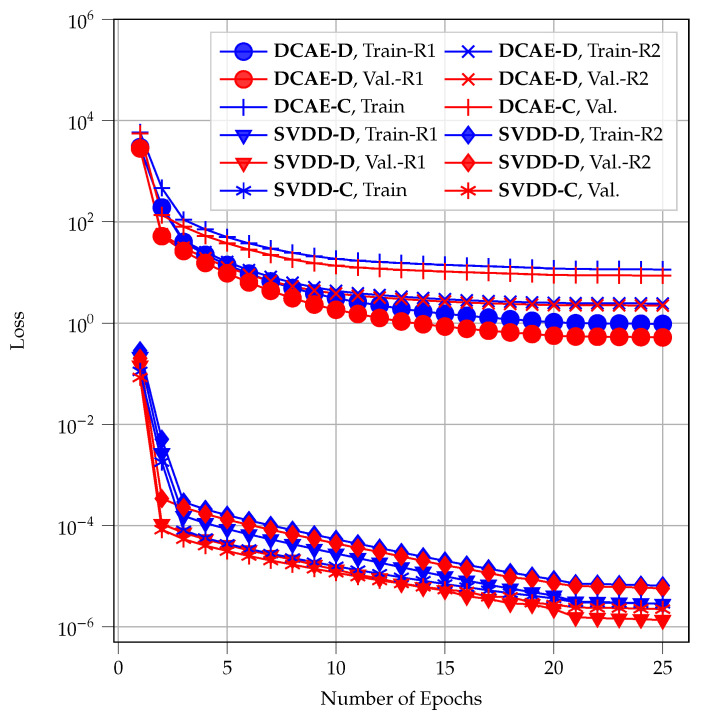
Learning curves of the DCAE used in **DCAE-D**, **DCAE-DP**, **DCAE-C**, and **DCAE-CP** and the Deep SVDD used in **SVDD-D** and **SVDD-C**. The learning curves for the training dataset (denoted as Train), the validation dataset (denoted as Val.), and two receivers, in the case of decentralized approaches (denoted as R1 and R2), are depicted.

**Figure 12 sensors-22-06547-f012:**
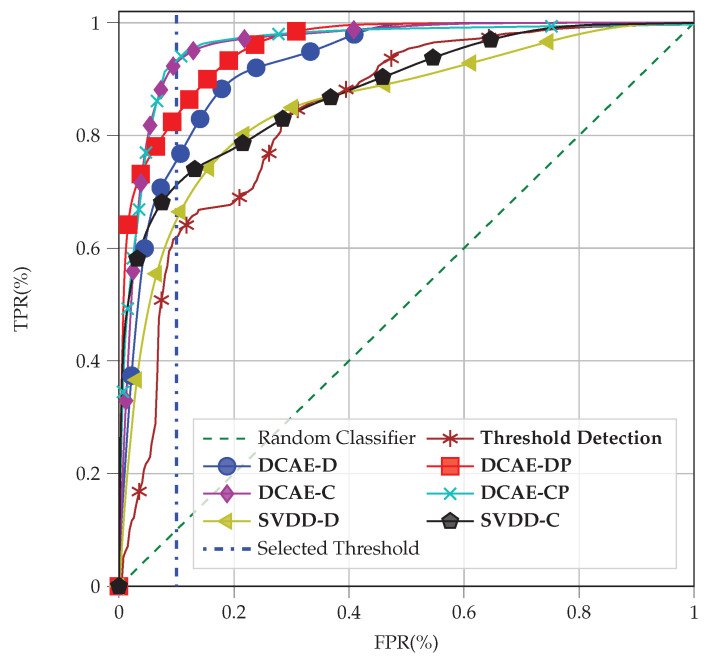
ROC curve of the methods with $N_{\mathrm{On}}$ = 300 in the two environments on average.

**Figure 13 sensors-22-06547-f013:**
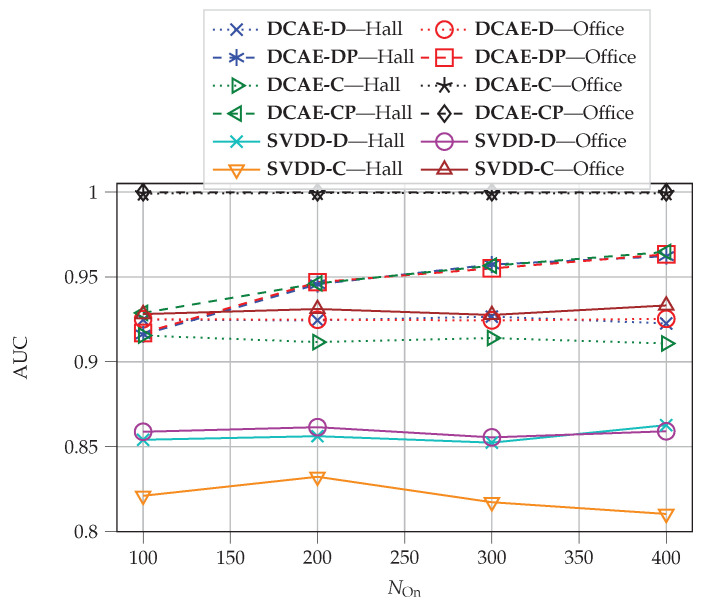
Impact of online batch size $N_{\mathrm{On}}$ on the AUC-ROC.

**Table 1 sensors-22-06547-t001:** A comparison of the relevant literature.

Reference-Year	System	Data	Environment	Remark
[3]-2007	DSSS	CIR	few dynamic elements	high misdetection rate in dynamic environments
[1]-2015	802.11n Wi-Fi	CSI	dynamic	using multiple receivers
[8]-2018	IoT	sensor data	dynamic	supervised ML algorithm
[6]-2021	OFDM-based	CSI	dynamic	semi-supervised DL algorithm
[7]-2021	COTS Wi-Fi	CSI	static	using a COTS Wi-Fi device as a detector
[9]-2021	IoT	packet length	dynamic	unsupervised ML algorithm

DSSS: direct sequence spread spectrum; in this table, the term “IoT” is used to indicate that the corresponding works utilized data from different networks such as WiFi, Zigbee, and Bluetooth.

**Table 2 sensors-22-06547-t002:** Measurement scenarios for the tamper-free default orientation.

Scenario	Description
**A**	a person sits in chair 1
**B**	same as A one hour later
**C**	a person walks in the area randomly
**D**	same as C one hour later
**E**	two persons walk in the area randomly
**F**	same as E one hour later
**G**	three persons walk in the area randomly
**H**	same as G one hour later

**Table 3 sensors-22-06547-t003:** Size of the CSI estimate datasets from the office and the hall environment.

Data Set	Office	Hall
**Training**	96,000	101,800
**Validation**	23,600	24,800
**Testing**	214,400	263,800

**Table 4 sensors-22-06547-t004:** Neuralnetwork parameters.

Description	Value
Optimizer	Adam
$N_{\mathrm{Off}}$	200
Weight Decay	10^−1^
Number of Epochs	20 + 5
Activation Function	ELU
Learning Rate	10^−5^ → 10^−6^

**Table 5 sensors-22-06547-t005:** Time complexity of the methods for each sample in the online phase. $N_{\mathrm{Off}}$ and *S* are the training batch size and number of subcarriers, respectively.

Method	No. of Multiplications	No. of Additions
**Threshold Detection**	≈$2SN_{\mathrm{Off}}^{✩}$	≈$4SN_{\mathrm{Off}}^{✩}$
**DCAE-D**	≈1978.75$N_{\mathrm{Off}}S$	≈1987.95$N_{\mathrm{Off}}S$
**DCAE-DP**	≈(1978.75 + $\frac{N_{\mathrm{On}}-1}{S}$)$N_{\mathrm{Off}}S$	≈1987.95$N_{\mathrm{Off}}S$
**DCAE-C**	≈1189.38$N_{\mathrm{Off}}S$	≈1193.94$N_{\mathrm{Off}}S$
**DCAE-CP**	≈(1189.38 + $\frac{N_{\mathrm{On}}-1}{S}$)$N_{\mathrm{Off}}S$	≈1193.94$N_{\mathrm{Off}}S$
**SVDD-D**	≈702.00$N_{\mathrm{Off}}S$	≈702.95$N_{\mathrm{Off}}S$
**SVDD-C**	≈351.00$N_{\mathrm{Off}}S$	≈352.44$N_{\mathrm{Off}}S$

**Table 6 sensors-22-06547-t006:** Performance of the methods on the measurement data (average over online batches).

Dataset	Method	AUC	Training Dur.	Test Dur.
Office	**Threshold Detection**	85.66%	-	3.92 s
Hall	**Threshold Detection**	83.53%	-	4.76 s
Office	**DCAE-D**	92.49%	329.50 s	16.99 s
Hall	**DCAE-D**	92.46%	334.40 s	17.14 s
Office	**DCAE-DP**	94.56%	337.34 s	42.08 s
Hall	**DCAE-DP**	94.53%	342.25 s	42.46 s
Office	**DCAE-C**	99.93%	237.65 s	12.95 s
Hall	**DCAE-C**	91.30%	247.54 s	16.08 s
Office	**DCAE-CP**	99.99%	244.54 s	21.33 s
Hall	**DCAE-CP**	94.91%	261.24 s	26.24 s
Office	**SVDD-D**	85.87%	242.63 s	11.59 s
Hall	**SVDD-D**	85.64%	246.99 s	11.79 s
Office	**SVDD-C**	93.00%	177.63 s	7.95 s
Hall	**SVDD-C**	82.03%	185.22 s	9.76 s

**Table 7 sensors-22-06547-t007:** Space complexity of methods.

Method	No. of Floating Point Elements
**Threshold Detection**	*S*$N_{\mathrm{Off}}^{✩}$R
**DCAE-D**	2901R
**DCAE-DP**	(2901 + $N_{\mathrm{On}}$)R
**DCAE-C**	3301
**DCAE-CP**	3301 + $N_{\mathrm{On}}$R
**SVDD-D**	1776R
**SVDD-C**	1976

## Data Availability

The dataset used in this work is available at https://github.com/isaac1369/Physical-Tamper-Attack-Detection.git (accessed on 20 July 2022).

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
