# Peer review of "DL-Based Physical Tamper Attack Detection in OFDM Systems with Multiple Receiver Antennas: A Performance–Complexity Trade-Off"

_sensors, 2022, doi:10.3390/s22176547_

Round 1

Reviewer 1 Report

This paper proposes and evaluates two DL-based approaches for detecting a physical tamper attack using CSI in an OFDM-based wireless communication system with multiple receiver antennas with respect to detection performance, time and space complexities. The experimental results show the correctness and effectiveness of the proposed method. There are some problems of theoretical and experimental analyses in this manuscript and it can be revised in the following aspects.

1. All mathematical expressions should be meticulously examined to avert some potential mistakes. For instance, how to choose the Threshold in Eq. 3, 8 and 10? How to select the R* in Eq. 14 and 16? The K under Eq.1 and the K in Eq. 5 should have the same meaning. The 2-norm in Eq. 4 should be equal to a scalar, not a vector.

2. The proposed methods are the extension of Ref. [6] and the application of Ref. [13]. Therefore, in the experimental section, the comparison with the state-of-the-art approaches are necessary.

Reviewer 2 Report

This paper, proposes to detect physical tampering attacks on wireless channels in the presence of Channel State Information (CSI) using two Deep Learning architectures: Deep Convolutional Autoencoder (DCAE) and Deep Support Vector Data Description (SVDD). In general, this paper is well written and has solid contributions in the dominance of knowledge, however, minor comments have arisen in reviewing the paper, which I summarise here.
1- The abstract states that this paper considers multiple receiving antennas. However, the assessment scenario uses a single receiver with 8 different antenna orientations. It is not clear to me if these antennas are active all the time or if they are changed during the simulation, as the evaluation results do not address the performance of the model in terms of antenna orientations. More importantly, can the proposed model be used for multiple receivers?
2- Which mobility model is used for the evaluation? If it is an empirical model, to what extent can the use of the known mobility models influence the results?
3- More details on the structure of the dataset are needed, as it is not clear how the 2D CNN is used to handle a 1D dataset.
